

# Larger than expected organic acid yields from the multi-generation oxidation of petrochemical alkenes

Baocong Zhao[1,2], Luxin Ren[1,2], Sihao Lin[1,2], Yongpeng Ji[1,2], Jiaxin Wang[1,2], Tao Ma[1,2], Yuemeng Ji[1,2], Taicheng An[1,2]

[1]Guangdong-Hong Kong-Macao Joint Laboratory for Contaminants Exposure and Health, Guangdong Key Laboratory of Environmental Catalysis and Health Risk Control, Institute of Environmental Health and Pollution Control, Guangdong University of Technology, Guangzhou 510006, China
[2]Guangzhou Key Laboratory of Environmental Catalysis and Pollution Control, Key Laboratory for City Cluster Environmental Safety and Green Development of the Ministry of Education, School of Environmental Science and
Engineering, Guangdong University of Technology, Guangzhou 510006, China

*Correspondence to*: Prof. Yuemeng Ji (jiym@gdut.edu.cn)

**Abstract.** Alkenes are primary pollutants in petrochemical source atmospheres, and their atmospheric chemistry is of great importance for tropospheric ozone and secondary organic aerosol formation. Hence, combining quantum chemical calculations and kinetic modelling, we investigated the oxidation mechanism and kinetics of 2-butene (BU), as one of the most important

alkenes, and its impact on the environment. The mechanism results show that OH addition is the dominant pathway for *cis*- and *trans*-isomers of BU, and then the corresponding OH-adducts are attacked by $O_2$ to produce peroxy radicals, which further react with NO to form acetaldehyde and hydroxyalkyl radicals. Different from the one adopted in current atmospheric models, addition of hydroxyalkyl radicals by $O_2$ and NO to form acetic acid proceeds with a smaller barrier than that for H-abstraction by $O_2$ to form acetaldehyde. A lifetime of less than a few hours (< 4 hours) for BU is estimated in the petrochemical regions.

Kinetic modelling demonstrates that oxidation of BU is predicted to yield significant amounts of organic acids (> 56%) in the petrochemical areas, larger than those are currently recognized, even in environments with low NO concentrations. Our results reveal that the OH-initiated oxidation of BU contributes importantly to organic acid budgets, particularly in the petrochemical regions, bridging the gap in organic acid budgets.

## 1 Introduction

Anthropogenic volatile organic compounds (AVOCs) have been associated with climate change, air quality, and environmental impacts via exposure to primary emissions and/or after their photochemical behaviors and multi-generation oxidation (Srivastava et al., 2022; Brook et al., 2010; Chen et al., 2024; Li et al., 2024). The latter leads to secondary air pollution, including secondary organic aerosol (SOA), tropospheric ozone, secondary organic acid, and so on (Peng et al., 2021; Yang et al., 2024; Tan et al., 2019). Automotive emissions of AVOCs have steadily decreased from efforts to control tailpipe

emissions in China, and as a result, other sources of AVOC emissions are growing in relative importance. Among them, petrochemical emissions are appreciable quantities with proportions for 7 - 26% of total AVOC emissions. Furthermore,



petrochemical emissions exhibit the largest potential of SOA formation (~23.7%) among all industrial emission processes (Wu and Xie, 2018). However, owing to the complex atmospheric chemistry of AVOCs from petrochemical emissions, their reaction mechanisms are uncharacterized, hindering the accurate assessment of their role in air quality and global climate.

Alkenes represent a significant proportion of AVOCs in petrochemical industrial areas (Guo et al., 2022b; Guo et al., 2022a; Henderson et al., 2010; Yang et al., 2023). A field observation revealed that alkenes accounted for $53.7 \pm 8.5\%$ of the total VOCs in the Lanzhou petrochemical area, with an average daily concentration of $82.3 \pm 13.1$ ppb (Yang et al., 2024). Previous studies have shown that multi-generation oxidation of alkenes is important to the formation of local free radicals, tropospheric ozone, and SOA (Wu et al., 2021; Wang et al., 2022; Yang et al., 2023; Tan et al., 2024). For example, the reaction of isoprene

with nitrate radicals ($NO_3$) produces the high nitrogen-containing monomers and dimers, leading to an estimated yield of organic aerosol mass of approximately $(5 \pm 2)$ % (Wu et al., 2021). Recent studies have pointed out that multi-generation oxidation of alkenes contribute to 8 - 20% of SOA mass (Lee et al., 2022) and about 89% of $O_3$ formation (Yang et al., 2024) in the petrochemical regions. Hence, the atmospheric chemistry of alkenes causes significantly secondary pollution to the petrochemical regions. Given the ubiquity of alkenes in petrochemical areas, understanding the multi-generation oxidation

mechanisms of anthropogenic alkenes is crucial for accurately predicting their impacts on air quality.

Although multi-generation oxidation of alkenes is believed to be important to SOA and $O_3$ formation, some studies have highlighted an unignorable source of the multi-generation oxidation for alkenes to secondary organic acids in the troposphere (Link et al., 2021; Srivastava et al., 2023; Wang et al., 2020; Isaacman-Vanwertz et al., 2018; Friedman and Farmer, 2018; Larsen et al., 2001). For example, a previous experimental study has revealed that organic acids obtained by the oxidation of

isoprene and α-pinene account for about 28% of the initial organic carbon (Link et al., 2021). However, a global chemistry-climate model simulation has shown that formic acid, an important class of organic acids in the atmosphere, was underestimated by 2 to 5 times relative to that of satellite observations (Franco et al., 2021). Therefore, current models still highly underestimate ambient concentrations of these acids, indicating that significant sources of organic acid in the atmosphere remain unidentified. Previous studies have identified several missing sources, including primarily from

combustion emissions, biogenic emissions, aqueous-phase chemistry of oxygenated VOCs, and photochemical reactions of alkenes (Paulot et al., 2011; Müller et al., 2019; Shaw et al., 2018; Chaliyakunnel et al., 2016; Franco et al., 2021; Link et al., 2021; Parandaman et al., 2018). Therefore, it is interesting to investigate the impact of their photochemistry on the formation of organic acids, to narrow the gap between observed and modelled organic acid concentrations, especially in typical petrochemical areas.

In the present study, we investigated the multi-generation oxidation mechanism and kinetics of 2-butene (BU) initiated by hydroxyl radical (OH) using a combination of quantum chemical calculations and kinetic modelling. BU is a representative alkene in the petrochemical regions due to its high abundance (Li et al., 2017; Ren et al., 2024; Wang et al., 2022; Zeng et al., 2022). Based on the mechanisms and kinetics, we also applied photochemical box model simulations to study the yields of organic acids from BU. The effects of OH and NO on the reaction mechanisms of BU were evaluated, and the implications of

organic acids formation were discussed.



## 2 Methods

All geometries of the reactants (Rs), pre-reactive complexes, transition states (TSs), intermediates (IMs) and products in this study were fully optimized at the M06-2X/6-311+G(2df,2p) level (Zhao and Truhlar, 2008). Harmonic vibrational frequencies were performed at the same level to verify the nature of transition state (NIMAG=1) and minimum (NIMAG=0), and to provide
zero-point vibrational energy (ZPVE), which is scaled by a factor of 0.967. Intrinsic reaction coordinate (IRC) calculations were carried out at the M06-2X/6-311+G(2df,2p) level to verify that each TS is connected to the desired reactants and products (Fukui, 1981). The single-point energy (SPE) calculations were further refined by the DLPNO-CCSD(T)/aug-cc-pVTZ level (Riplinger et al., 2013) with normal pair natural orbital (NormalPNO) criteria (Liakos et al., 2015) to yield more accurate energetics. $T_1$ diagnostic values in the DLPNO-CCSD(T) calculations for the IMs and TSs involved in the key reaction
pathways were checked for multi-reference character. The $T_1$ diagnostic values for all checked important species in this work are lower than the threshold value of 0.045, indicating the reliability of applied single reference methods. In all cases, the energies were calculated relative to the corresponding reactants including ZPVE corrections. $\Delta E_a^{\#}$ is defined as the activation energy ($\Delta E_a^{\#} = E_{TS} - E_{Reactant}$), while $\Delta E_r$ is defined as the reaction energy ($\Delta E_r = E_{Product} - E_{Reactant}$). All above calculations were performed within Gaussian 09 and ORCA 5.0.0 program (Frisch, 2009; Neese, 2012). The Multiwfn program and Visual
Molecular Dynamics (VMD) were utilized to analyze and visualize the molecular orbitals of relevant species (Humphrey et al., 1996; Lu and Chen, 2012).

The rate constants for the reactions with TSs were calculated using the variational transition state theory (VTST) along with one-dimensional asymmetric Eckart tunneling correction (Bao and Truhlar, 2017; Eckart, 1930). Besides, the rate constants for the barrierless reactions were calculated by employing the variable-reaction-coordinate variational transition state theory
(VRC-VTST) (Bao and Truhlar, 2017). For pathways involving multiple conformers, the rate constants were calculated using multi-conformer transition state theory (MC-TST) (Møller et al., 2016), incorporating data for all conformers obtained from the Molclus program (Lu, 2020). All the kinetics calculations were performed with the KiSThelP 2021 and Polyrate 2017-C programs (Canneaux et al., 2014; Zheng, 2018).

A box-model was used to investigate the formation of organic acids, built using AtChem 2 Program (Sommariva et al., 2020;
http:https://atchem.leeds.ac.uk/, last access: 1 July 2025) with a chemical mechanism taken from the Master Chemical Mechanism (MCM v3.3.1) (Jenkin et al., 1997; Saunders et al., 2003; http://mcm.leeds.ac.uk/MCM, last access: 1 July 2025). The box model was constrained by the initial concentrations of NO, $NO_2$, BU, and OH as listed in Table S12 (Yang et al., 2024; Yang et al., 2023). The concentration of OH ([OH]) was kept constant throughout the simulation over a two-hour period. More details about the box model simulation are shown in the Supplement.



## 3 Results and discussion

### 3.1 OH-Initiated reactions of *cis*- and *trans*-BU

In the atmosphere, there exists two stable isomers of BU, i.e., *cis*-2-butene (*cis*-BU) and *trans*-2-butene (*trans*-BU), which are proven to be ubiquitous and difficult to interconvert (Tuazon et al., 1998; Wang et al., 2022; Mo et al., 2022). Hence, to systematically assess the photochemistry of BU, we considered the OH-initiated reactions of *cis*-BU and *trans*-BU. Figure S1 displays the optimization of geometries for all single points (SPs) involved in these two reactions at the M06-2X/6-311+G(2df,2p) level. For comparison, other levels, including the B3LYP/, MPW1PW91/, and $\omega$B97X-D/6-311+G(2df,2p) levels, were performed to calculate the geometries (Figure S2). The structural parameters of all SPs obtained by the four levels are similar, with the largest discrepancies of less than 0.01 Å in bond lengths and 1.70° in bond angles. Hence, the M06-2X level of theory can accurately describe the geometrical information of OH-initiated reactions of *cis*-BU and *trans*-BU.

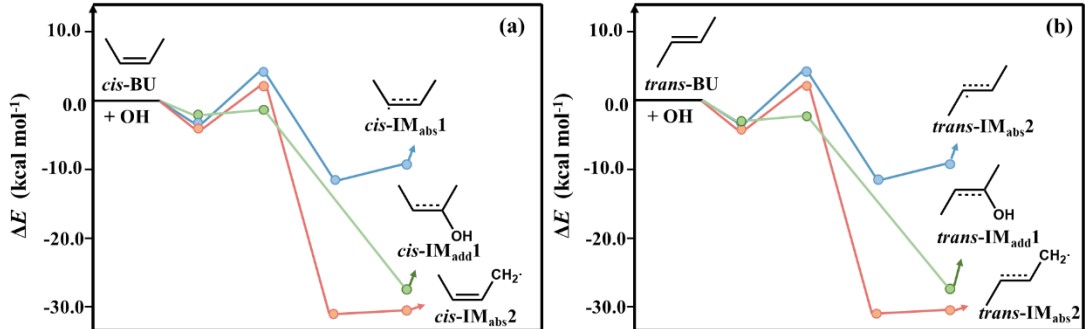

**Figure 1: Potential energy surfaces (PESs) for the OH-initiated reactions of (a) *cis*-BU and (b) *trans*-BU with OH (unit: kcal mol$^{-1}$).**

The potential energy surfaces (PESs) for all possible pathways of the OH-initiated reactions of *cis*- and *trans*-BU are presented in Figure 1. There are two kinds of pathways, i.e., OH-addition (R$_{add}$) and H-abstraction (R$_{abs}$). Each pathway has a pre-reactive complex prior to the corresponding TS, which is more stable than the corresponding reactants. As shown in Figure S1, the $C_{2v}$ symmetry of *cis*-/*trans*-BU suggests one OH-addition (*cis*-/*trans*-R$_{add}$) and two H-abstraction (*cis*-/*trans*-R$_{abs}$) pathways in the *cis*-/*trans*-BU reaction systems. For *cis*-BU, the pathway of H-abstraction from -CH$_3$ (*cis*-R$_{abs}$1) possesses a lower $\Delta E_a^{\#}$ value of 2.36 kcal mol$^{-1}$ and a more negative exothermicity of -30.96 kcal mol$^{-1}$ than H-abstraction from -CH= group (*cis*-R$_{abs}$2). It is attributed to the smaller bond dissociation energy of C-H bond ($D_{298}^{0}$(C-H)) at the -CH$_3$ groups relative to that at the -CH= groups in *cis*-BU (Table S1). It suggests that the H-abstraction from the -CH$_3$ group is more favorable than that from the -CH= group. However, OH-addition to C=C double bond (*cis*-R$_{add}$1) proceeds via a negative $\Delta E_a^{\#}$ value of -0.97 kcal mol$^{-1}$, which is at least 3 kcal mol$^{-1}$ smaller than those of H-abstraction pathways, indicating favorable formation of the OH-adduct intermediate (*cis*-IM$_{add}$1). Similarly, OH-addition to *trans*-BU (*trans*-R$_{add}$1) is also a dominant pathway, but it possesses a more negative $\Delta E_a^{\#}$ value of -1.33 kcal mol$^{-1}$ than the *cis*-R$_{add}$1 pathway. The interaction region indicator listed in Figure S3 reveals that OH-addition to the C=C bond of *cis*-BU exhibits a stronger steric hindrance compared to *trans*-BU, attributed to





the van der Waals and steric hindrance interactions between the two -CH$_3$ groups in *cis*-TS$_{add}$1. The larger Mayer bond order of the forming C-O bond in *trans*-TS$_{add}$1 (0.251) relative to that in *cis*-TS (0.192) suggests stronger electronic interactions in the *trans*-TS$_{add}$1. This difference, combined with steric hindrance, primarily influences the reactivity of the C=C bond in both *cis*-BU and *trans*-BU, leading to the formation of stable OH-adduct intermediates (*cis*- and *trans*-IM$_{add}$1).

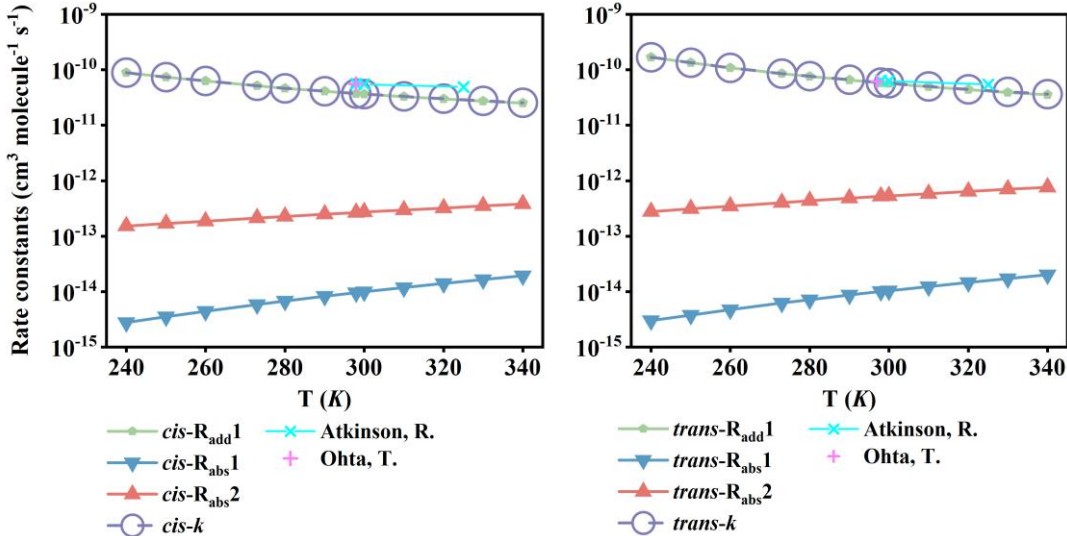


**Figure 2: The rate constants (cm$^3$ molecule$^{-1}$ s$^{-1}$) of (a) *cis*-BU and (b) *trans*-BU with OH against the temperature range of 240-340 K. The experimental rate constants are from Atkinson, 2000 and Ohta, 1984 respectively.**

The rate constant of each pathway and the total rate constants ($k_{total}$) of the OH-initial reactions of *cis*- and *trans*-BU were calculated and are summarized in Figure 2, along with comparisons with the available experimental data (Atkinson and Arey, 2003; Sims et al., 1994; Ohta, 1984). The total rate constant of each reaction system is obtained by the sum of calculated rate constants for all pathways. There is a negative correlation between rate constants and temperatures in the temperature range of 240 - 340 K, attributable to the presence of the pre-reactive complexes. Our calculated rate constants at the DLPNO/aug-cc-pVTZ//M06-2X/6-311+G(2df,2p) level also compare favorably with the available experimental data. For example, the rate constant of the *trans*-BU + OH reaction at 298 K is 5.85 × 10$^{-11}$ cm$^3$ molecule$^{-1}$ s$^{-1}$, in lines with the experimental values of (5.40 ± 0.02) × 10$^{-11}$ cm$^3$ molecule$^{-1}$ s$^{-1}$ reported by Sims et al., 1994 and (6.09 ± 0.3) × 10$^{-11}$ obtained by Ohta, 1984. Hence, the DLPNO//M06-2X method provides a reliable description for the kinetics of the OH-initial reactions of *cis*-BU and *trans*-BU.

Tables S4-S5 list the temperature dependences of the branching ratios ($\Gamma$) over the temperature range from 240 to 340 K. For *cis*-BU, the rate constant of OH-addition pathway is at least two orders of magnitude higher than those of the corresponding H-abstraction pathways. The contribution of OH-addition pathway to the total rate constant is more than 99% in the whole measured temperature ranges. Similarly, for *trans*-BU, the OH-addition pathway is also of major importance. Arrhenius expressions are derived to be $k_{total}$(*cis*-BU + OH) = 1.23 × 10$^{-12}$ exp (1021.32/T) and $k_{total}$(*trans*-BU + OH) = 8.90 × 10$^{-13}$ exp






(1251.48/T) over the temperature range of 240-340 K. From the calculated Arrhenius expressions, the activation energies of the *cis-* and *trans*-BU reaction systems are deduced to be -2.03 and -2.49 kcal mol$^{-1}$, respectively. These negative activation energy values indicate that the OH-initiated reactions of *cis-* and *trans*-BU are kinetically favored in the troposphere, to rapidly form the OH-adduct intermediates.

## 3.2 Subsequent reactions of the OH-adduct intermediates

The OH-adduct intermediates, i.e., *cis-* and *trans*-IM$_{add}$1, proceed via three competitive pathways including reaction with O$_2$ (R3), isomerization (R4), and decomposition (R5). A Schematic PES presented in Figure S4 reveals that the TSs of *cis-* and *trans*-R3 are not identified, but there are TSs for R4 and R5 pathways with the $\Delta E_a^{\#}$ values more than 30.0 kcal mol$^{-1}$. The calculated rate constants of R4 and R5 pathways are 7.3 × 10$^{-13}$ and 2.6 × 10$^{-11}$ s$^{-1}$ in the *cis-* and *trans*-BU reaction systems (Table S6), which are 18-20 orders of magnitude smaller than that of R3 (6.0 × 10$^{-11}$ cm$^3$ molecule$^{-1}$ s$^{-1}$, corresponding to an equivalent first-order rate constant of 3.0 × 10$^7$ s$^{-1}$). Considering the branching ratios among the three pathways, about 99% of both *cis-* and *trans*-IM$_{add}$1 react with O$_2$ to form peroxy radicals (*cis-/trans*-RO$_2$), which further propagate the oxidation.

For *cis*-RO$_2$, attacked by NO (*cis*-R6) or HO$_2$ radical (*cis*-R7) forms the peroxy nitrite (*cis*-RO$_2$NO) or the hydroperoxide (*cis*-ROOH). Alternatively, there exists the autoxidation of *cis*-RO$_2$, which proceeds via two-step reactions, i.e., a H-shift reaction (*cis*-R8) followed by an O$_2$-addition (*cis*-R9), with the high $\Delta E_a^{\#}$ value and the small rate constant (Figures S7-S8). Similarly, the reaction with HO$_2$ radical also proceeds via a TS with the $\Delta E_a^{\#}$ value of 4.20 kcal mol$^{-1}$. However, the association reaction of *cis*-RO$_2$ with NO is barrierless and exothermic (Figure 3), and the corresponding equivalent first-order rate constant is 1.1 s$^{-1}$, which is at least five orders of magnitude higher than those of *cis*-R7 and *cis*-R8 pathways (Tables S7-S9). As shown in Figure S8, the reaction of *cis*-RO$_2$ with HO$_2$ is competitive in the troposphere only if HO$_2$ concentration exceeds 40 ppt, which is the maximum atmospheric concentration. It implies that the *cis*-RO$_2$NO is the dominant product from *cis*-RO$_2$. Subsequently, there are three reaction pathways of *cis*-RO$_2$NO, i.e., NO$_2$-elimination (*cis*-R10), intramolecular isomerization (*cis*-R11), and dissociation (*cis*-R12). The $\Delta E_a^{\#}$ values of *cis*-R11 and *cis*-R12 are 54.23 and 38.35 kcal mol$^{-1}$ (Figure 3), respectively, which are 26.36 kcal mol$^{-1}$ larger than that of *cis*-R10. It indicates that the formation of organic nitrates (*cis*-ON), acetoin (CH$_3$CH(OH)C(O)CH$_3$), and HONO is of minor importance. As shown in Figure 3, the favorably produced alkoxy radical (*cis*-RO) via *cis*-R10 then undergoes dissociation (*cis*-R13), isomerization (*cis*-R14), and H-abstraction (*cis*-R15) yield acetaldehyde (CH$_3$CHO) and hydroxyalkyl radicals (CH$_3$CHOH), CH$_3$CH(OH)CH(OH)CH$_2$ radical, and acetoin (CH$_3$C(=O)CH(OH)CH$_3$), respectively. The *cis*-R13 pathway possesses a smaller $\Delta E_a^{\#}$ value of 6.51 kcal mol$^{-1}$ and a larger rate constant of 1.10 × 10$^8$ s$^{-1}$ at 298 K relative to the *cis*-R14 and *cis*-R15 pathways (Table S10), indicating a major importance to form CH$_3$CHO and CH$_3$CHOH.

Similarly, as shown in Figure S5, the subsequent reactions of *trans*-RO$_2$ involve three essential steps: (i) the association with NO to form *trans*-RO$_2$NO, (ii) the NO$_2$-elimination of *trans*-RO$_2$NO to produce *trans*-RO, and (iii) the dissociation of *trans*-RO to yield CH$_3$CHO and CH$_3$CHOH radical. However, the differences between the subsequent reactions of *cis*-IM$_{add}$1 and



$trans$-$IM_{add}1$ are reflected in the lower reaction energy barriers and the larger rate constants of $trans$-$IM_{add}1$. For example, the $\Delta E_a^{\#}$ value of $trans$-R12 pathway is 11.99 kcal mol$^{-1}$, which is 2.6 kcal mol$^{-1}$ lower than that of $cis$-R12 pathway, and the corresponding rate constant of $trans$-R12 pathway is six times larger than that of $cis$-R12 pathway. It is attributed to a stronger steric hindrance in subsequent reactions of $cis$-$IM_{add}1$ relative to $trans$-$IM_{add}1$. Therefore, CH$_3$CHO and CH$_3$CHOH radicals are more rapidly produced from $trans$-BU relative to $cis$-BU in the atmosphere.

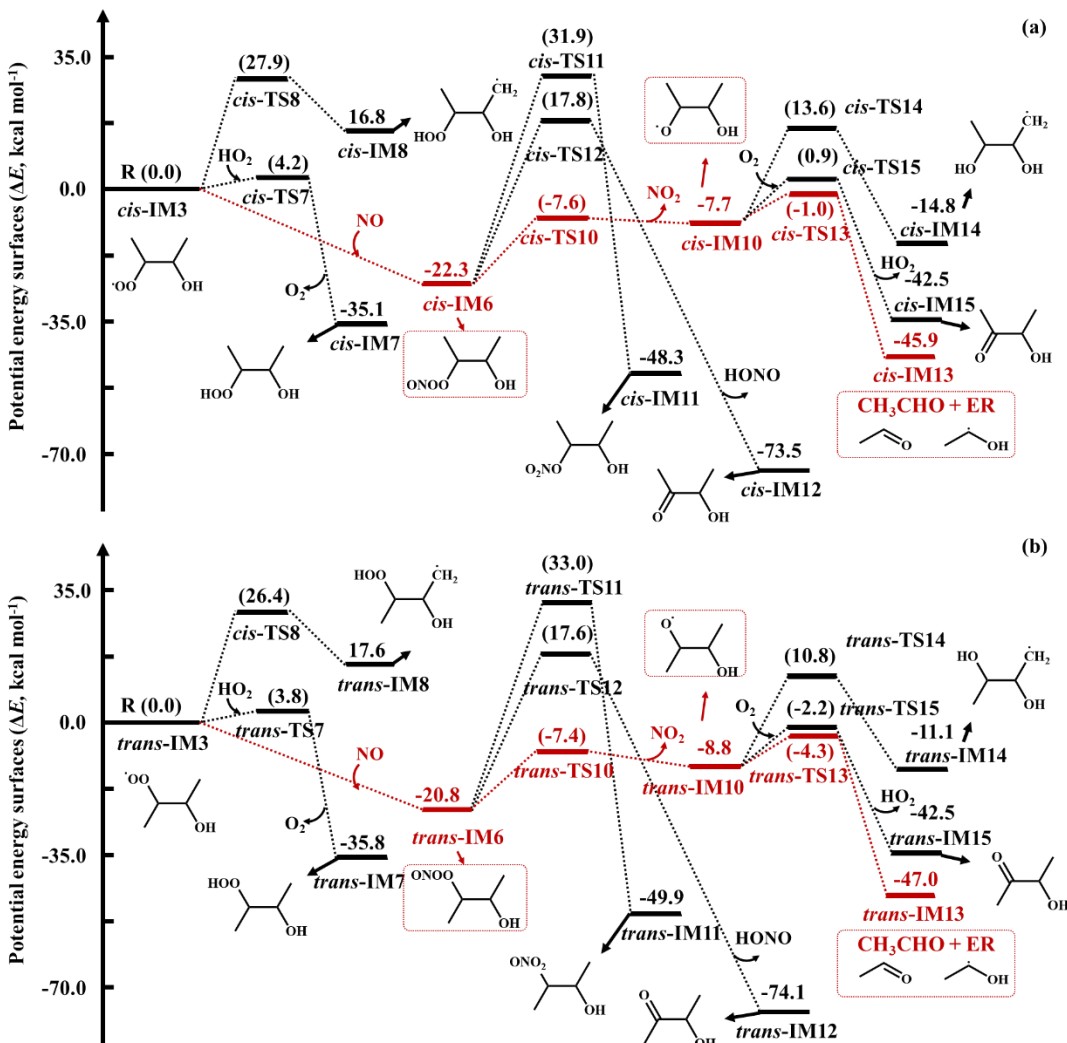

**Figure 3: PESs of the reaction of (a) $cis$-IM3 and (b) $trans$-IM3. The number denotes the $\Delta E_a^{\#}$ and $\Delta E_r$ for each reaction step.**

### 3.3 The fate of BU

According to the previous study (Parandaman et al., 2018), CH$_3$CHOH·radical undergoes the H-abstraction pathway (R16) to form CH$_3$CHO and HO$_2$ radical. However, the corresponding $\Delta E_a^{\#}$ and $\Delta E_r$ values are calculated to be 11.27 kcal mol$^{-1}$ and





11.69 kcal mol$^{-1}$, respectively, and the pathway of O$_2$-addition to CH$_3$CHOH radical (R17) is barrierless and largely exothermic (-34.11 kcal mol$^{-1}$) to yield peroxy radical (ER-O$_2$) (Figure 4). More importantly, the subsequent NO-association of ER-O$_2$ (R18) is also barrierless and largely exothermic (-24.16 kcal mol$^{-1}$). Therefore, peroxy nitrite (ER-O$_2$NO) is the dominant product of the subsequent reaction of CH$_3$CHOH via R17 and R18 rather than CH$_3$CHO and HO$_2$ radical via R16. However, subsequent decomposition of ER-O$_2$NO (R19) possesses a relatively high $\Delta E_a^{\#}$ value of 13.69 kcal mol$^{-1}$, which can be

overcome because the excess energy from ER-O$_2$NO formation allows the proceeding decomposition leading to the formation of alkoxy radical (ER-O) and NO$_2$. ER-O reacts with O$_2$ (R20) to produce acetic acid (CH$_3$COOH) and HO$_2$ radical, with the $\Delta E_a^{\#}$ and $\Delta E_r$ values of 5.40 and -48.13 kcal mol$^{-1}$, respectively. The branching ratios between H-abstraction to form CH$_3$CHO and HO$_2$ radical and O$_2$-addition to yield CH$_3$COOH and HO$_2$ radical reveals that the formation of CH$_3$COOH is major importance (> 69%) from the multi-generation oxidation of BU (Figure 4), which requires to further assess the contribution of

BU to organic acids in the atmosphere. Furthermore, we predict a minor pathway via RO$_2$ + NO reaction to form organic nitrate, consistent with the work by Muthuramu et al., 1993 for a small yield for the formation of organic nitrate from *cis*-BU (3.7±0.9%).



**Figure 4: Schematic representation of the preferred pathways of the *cis*-BU + OH reactions leading to formation of acetaldehyde**
**(CH$_3$CHO) and acetic acid (CH$_3$COOH). Values of branching ratio are shown in black.**

In addition, we also evaluate the tropospheric lifetimes ($\tau$) for *cis*-BU and *trans*-BU at different [OH] levels. The gas-phase OH oxidation lifetimes were estimated using $\tau = 1/(k_{total}[OH])$, where $k_{total}$ and [OH] are the total rate constant and OH concentration, respectively. As shown in Table S11, at the urban-level with [OH] = 1 × 10$^6$ molecule cm$^{-3}$, the $\tau$ values of *cis*-BU and *trans*-BU are 7.43 and 4.75 hours at 298 K, respectively, which are longer than those calculated in the petrochemical

regions with [OH] of 2 × 10$^6$ molecule cm$^{-3}$. The short $\tau$ value of BUs in the petrochemical regions indicate that they are more



readily oxidized, implying a more significant environmental impact in these regions compared to urban regions. The $\tau$ values of *cis*- and *trans*-BU are further shortened to be 2.70 and 1.62 hours as the temperature drops to 273 K in the petrochemical regions. Given that organic acids are more conducive to atmospheric new particle formation and growth under low-temperature conditions (Peng et al., 2021), it is essential to focus on the contribution of the BU oxidation in the petrochemical regions to

the formation of organic acids at low temperatures.

### 3.4 Rate of CH$_3$COOH formation

To evaluate the impact of our established mechanisms on the formation of organic acids in the petrochemical regions, a box model simulation was performed to quantify the production rate ($r_{CH3COOH}$) and yield ($Y_{CH3COOH}$) of CH$_3$COOH by using the MCM v3.3.1 coupled with the AtChem-2 box model. For comparison, the corresponding simulation under the traditional

mechanism of CH$_3$COOH formation in MCM v3.3.1 was also carried out. All the simulations were conducted using the measured mixing ratios of BUs, OH, and NO in a typical petrochemical industrial region, and all the settings were posted in Table S12 (Yang et al., 2024). It is evident from Figure 5 that the $r_{CH3COOH}$ values corrected with our proposed mechanism exhibit a significant increase of more than ten times under typical petrochemical conditions, where the concentrations of [OH] and NO ([NO]) are $2 \times 10^6$ molecule cm$^{-3}$ and 10 ppb ($2.5 \times 10^{11}$ molecule cm$^{-3}$), respectively. Even when [OH] and [NO]

decrease, the $r_{CH3COOH}$ values are also increased by more than five times. Large production rates of CH$_3$COOH correspond to the high $Y_{CH3COOH}$ values, for example, the $Y_{CH3COOH}$ value in *cis*-BU + OH reaction with 57% is more than ten times higher than that without our proposed mechanism under the typical petrochemical conditions. Hence, the photooxidation of *cis*-/*trans*-BU corrected by our proposed mechanisms has a significant impact on the formation of organic acids in the petrochemical source regions, with particularly pronounced effects on the formation of small-molecule gaseous organic acids.

To further assess the atmospheric regions where the photochemistry of BU will have significance, we also calculated the $r_{CH3COOH}$ values under the varying [OH] and [NO] corresponding to the atmospheric conditions (Figure 5) (Tan et al., 2019). In a high [NO] condition of $4.9 \times 10^{10}$ molecule cm$^{-3}$ with the [OH] of $6 \times 10^6$ molecule cm$^{-3}$, the $r_{CH3COOH}$ values are 0.51 and 0.47 ppb h$^{-1}$ in *cis*-BU + OH and *trans*-BU + OH reactions, which are at least 55% larger than those with the [OH] of $8 \times 10^5$ molecule cm$^{-3}$, respectively. Even in a low [NO] condition of $2.5 \times 10^9$ molecule cm$^{-3}$, the $r_{CH3COOH}$ values in two reactions are

also significantly regulated by the atmospheric [OH]. However, there is little effect of [NO] on the $r_{CH3COOH}$ values. For example, at the same [OH] of $2 \times 10^6$ molecule cm$^{-3}$, the $r_{CH3COOH}$ value in *cis*-BU + OH reaction is 0.49 ppb h$^{-1}$ at the [NO] of $5.0 \times 10^{11}$ molecule cm$^{-3}$, while in the [NO] of $2.5 \times 10^9$ molecule cm$^{-3}$, the $r_{CH3COOH}$ value is only decreased by 7%. Combined with the mechanism results, the yield of organic acid from the BU + OH reaction is regulated by the atmospheric OH concentration. Therefore, under both typical petrochemical source region conditions and general atmospheric conditions,

*cis*-BU + OH and *trans*-BU + OH reactions can form CH$_3$COOH through the subsequent oxidation of CH$_3$CHOH radicals, exhibiting unexpectedly high formation rates during daylight hours.





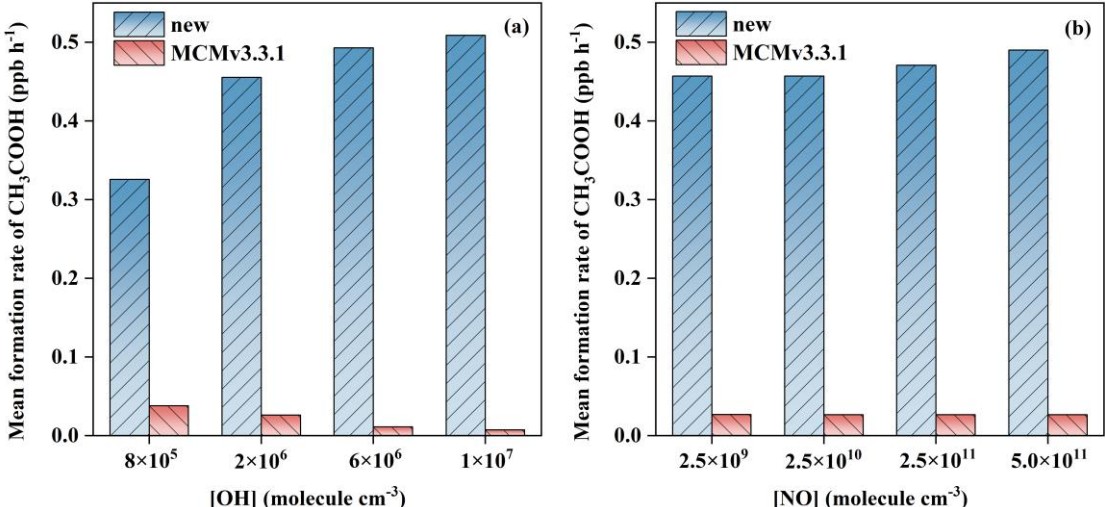

**Figure 5: Mean formation rate of CH3COOH (unit: ppb h⁻¹) from the *cis*-BU + OH reactions as a function of (a) OH concentration ([OH]) and (b) NO concentration ([NO]) under the petrochemical region conditions. The comparison is made between the MCMv3.3.1 and the new mechanism.**

## 4 Conclusions and atmospheric implications

AVOCs have profound impacts on air quality, human health and climate, and BU is the major AVOCs emitted from the petrochemical regions. Hence, from combined quantum chemical calculations and photochemical box model simulations, this study provides a systematic insight into the multi-generation oxidation mechanisms, kinetics, and atmospheric fate of BU and its contribution to the formation of organic acid. The initial reactions of both the *cis*- and *trans*-isomers of BU readily involve OH addition to the C=C double bond, yielding the corresponding OH-adducts. Subsequent reactions proceed via $O_2$ addition, NO-association, $NO_2$-elimination, and further decomposition to produce $CH_3CHO$ and $CH_3CHOH$ radical. $CH_3CHOH$ radical is converted into $CH_3COOH$ and $HO_2$ radical, facilitated by $O_2$ and NO through the pathways of $O_2$ addition, NO-association, $NO_2$-elimination, and H-abstraction. Our mechanism highlights that the rapid and irreversible reaction of $CH_3CHOH\cdot$ with $O_2$ and NO is a key mechanistic step in the formation of $CH_3COOH$, which is one of the most favorable products, with a yield of 57%. However, previous experimental studies on the OH-initiated oxidation of BU obtained a low yield of $CH_3COOH$ (< 10 %) (Atkinson, 1997; Muthuramu et al., 1993). Given the rapid partitioning of gas $CH_3COOH$ into the liquid phase in the experimental environment (Franco et al., 2021), the measured yield likely corresponds to the lower experimental limit.

This study further reveals that the multi-generation oxidation of BU in the petrochemical environment, may be particularly important for organic acid formation. Using our calculated kinetics data, we estimate a lifetime of less than 4 hours for BU in the petrochemical environment, indicating that BU are rapidly oxidized following the local emission. By adding our established mechanisms to the photochemical box model simulations, our calculated formation rates and yields of $CH_3COOH$ are at least 10 times higher than those from the traditional mechanism. Besides, the impact of OH concentrations on the formation rate of



CH₃COOH is greater than that of NO concentrations. That is, the formation rate of $CH_3COOH$ rapidly drops, as OH
concentrations decrease. This highlights the dominant role of OH radicals in the initial oxidation steps of multi-step oxidation
processes. With the increasing oxidative capacity and decreasing nitrogen oxides ($NO_x$) abundance in polluted areas (Newland
et al., 2021), further study is needed to explore the impacts of these factors on the multi-generation oxidation reactions of
alkenes.

Contrary to the consensus that the complex multi-generation oxidation of alkenes primarily contributes to the formation of
low-volatility products, we discovered that the multi-generation oxidation of BU can unexpectedly contribute to the formation
of organic acids. The unexpected production of organic acids can not only help to explain the missing source of organic acids
but also affect the acidity of atmospheric precipitation, especially for the petrochemical region atmosphere. In the atmosphere,
we predict that the multi-generation oxidation of alkenes during the daytime will produces $CH_3COOH$ at a rate of 10.4 ppb h$^-$
$^1$ (Figure S10), which can be comparable with the simulated source from the photochemical reaction of biogenic alkenes
(Paulot et al., 2011). Therefore, the mechanism for the formation of organic acids from BU isomers established by this study
is more significant than previously recognized, particularly in the petrochemical regions with the high emissions of BU isomers.
Further investigation is warranted into the multi-generation oxidation of AVOCs, as well as their impacts on the formation of
organic acids and the environment.

**Data availability.** The data are accessible by contacting the corresponding author (jiym@gdut.edu.cn).

**Supplement.** The following information is provided in the Supplement: the comparison of the geometries of *cis-* and *trans-*
isomers of BU calculated at the M06-2X/6-311+G(2df,2p) level of theory and other levels including B3LYP/6-311+G(2df,2p),
MPW1PW91/6-311+G(2df,2p), and $\omega$B97X-D/6-311+G(2df,2p); the interaction region indicator analyses of of *cis-*TS$_{add}$1 and
*trans-* TS$_{add}$1; rate constants of each elementary pathway involved in the reaction of *cis-*BU + OH and *trans-*BU + OH; PESs
for the subsequent reactions of *cis-* and *trans-*isomers of OH adduct, involving the bimolecular reactions with $O_2$, NO, and



HO₂, and the unimolecular reactions of isomerization and dissociation; geometries of all stationary points; mean formation rate of CH₃COOH from the *trans*-BU + OH reactions; parameter settings and simulation results of the AtChem 2 model.

**Author contributions.** YMJ and BCZ designed the research; YMJ, BCZ, LXR and SHL performed the research; BCZ, YPJ, LXR, JXW, TM and TCA analyzed the data; YMJ and BCZ wrote the paper; LXR, JXW, YPJ, TM and TCA reviewed and edited the paper.

**Competing interests.** The contact author has declared that neither they nor their co-authors have any competing interests.

**Financial support.** This work was financially supported by National Natural Science Foundation of China (42020104001 and 42077189), Guangdong Basic and Applied Basic Research Foundation (2025A1515011379), Technology Elite Navigation Project of Guangzhou (2025A04J7038), and Guang-dong Provincial Key R&D Program (2022-GDUT-A0007).

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
