# Peer review of "Larger than expected organic acid yields from the multi-generation oxidation of petrochemical alkenes"

_EGUsphere, 2025_

## Author Comment (AC3)

- 1 Response to Referees
- 2 We sincerely thank the valuable comments and suggestions from the referees. We
- 3 extensively revised the manuscript according to the referees' comments. The responses
- 4 to both referees are included in the attached document.

- 6 Referee #1
- 7 This is an interesting work that investigates the multi-generation OH-initiated oxidation
- 8 of 2-butene and its impact on the formation of organic acids in the petrochemical
- 9 regions. Mainly it finds that larger than expected organic acid yields from the multi-
- 10 generation oxidation of 2-butene in the petrochemical regions, originated from a novel
- pathway for acetic acid formation. It combines quantum calculation and photochemical
- box model simulation to deliver the critical thermodynamic and kinetic parameters for
- 13 atmospheric models and advances our understanding of petrochemical-alkene
- chemistry. The manuscript is well organized and clearly written; its topic is timely and
- 15 appropriate for publication.
- Response: We appreciate referee's suggestive comments.

17

- 18 1. Line 62: It would be helpful if the authors provide the atmospheric concentration of
- 19 *2-butene to illustrate that 2-butene is abundance in petrochemical regions.*
- 20 Response: According to the referee's helpful suggestion, the atmospheric concentration
- 21 of 2-butene was supplied to elaborate that 2-butene is abundant in petrochemical
- 22 regions in the revised manuscript: "BU is a representative alkene in the
- petrochemical regions, with the concentrations range from 0.5 ppb to 11.4 ppb."
- 24 (please see line 65).

- 26 2. Lines 85–87: The manuscript states that multi-conformer transition-state theory
- 27 (MC-TST) was used for "pathways involving multiple conformers," but it is not clear
- 28 how the authors identified which reaction channels possess multiple conformers. A brief
- 29 description of the conformational search protocol and the energetic or structural
- 30 criteria used to retain conformers (e.g., energy cut-off, rotational barriers, symmetry
- 31 considerations) should be added to the Methods section. Besides, Line 136:
- 32 *DLPNO/M06-2X* is not fully introduced anywhere.
- Response: Thank you for the referee's comment. We have added a brief description
- 34 about the MC-TST method in the "Method" of the revised manuscript: "The

35 combination of single-conformer and multi-conformers approximation was 36 adopted to investigate the reaction mechanism in a cost-effective way of 37 considering the effect of multiple conformers of the reactants and TSs. Based on 38 the single-conformer calculations, the effect of multiple conformers was 39 considered for the crucial reaction step of H-shift reaction of RO2. The systematic 40 structure scanning method was employed to produce the conformers by the 41 Molclus program (Lu, 2020). The systematic structure scanning was performed by 42 regularly rotating the dihedral angle that determines the conformers of the target 43 molecules. For the target molecules, it yielded 4-256 conformers depending on the 44 complexity of the system. Initial geometry optimizations and single-point 45 electronic energy calculations were performed at the B3LYP/6-31+G(d) level. 46 Subsequently, conformers with electronic energies within 2.0 kcal mol-1 relative to 47 the lowest-energy conformer were further considered for geometry optimization 48 at the M06-2X/6-311+G(2df,2p) level. On the lowest electronic energy R, 49 intermediate, TS, and product geometries at the M06-2X/6-311+G(2df,2p) level, 50 single-point calculation at the DLPNO-CCSD(T)/aug-cc-pVTZ level was carried 51 out.". (please see lines 85-95) 52 And we clarified the notation from "DLPNO//M06-2X" to "DLPNO-CCSD(T)/aug-53 cc-pVTZ//M06-2X/6-311+g(2df,2p)". (please see lines 150-151)

3. It would be valuable to elaborate the reasons why the temperature range of 240-340

K is selected to calculate the kinetics. Does the negative correlation for the rate

57 constant refer to the total k value? Since other values actually show a positive

58 correlation, it would be helpful to elaborate on this further.

54

56

65

66

67

68

Response: According to the referee's suggestion, the reasons why the temperature range

of 240-340 K is selected have been added in the revised manuscript: "The rate

61 constants were calculated over the temperature range of 240-340 K, considered

from the surface of the earth to the lower troposphere." (please see lines 143-144).

Yes, the negative correlation refers to the total k value ( $k_{\text{total}}$ ). For clarify, we have

revised the description from "rate constants" to "ktotal" in the revised manuscript:

"There is a negative correlation between  $k_{\text{total}}$  and temperatures over 240 - 340

**K...**". (please see line 145). In addition, based on the referee #2's comment, we also

found that the OH-addition pathway, as the dominant pathway, exhibits a negative

temperature effect, since it has a submerged energy barrier than the corresponding

- reactants; while H-abstraction pathways show a positive temperature dependence, since
- they do not have submerged energy barriers. Hence, we have elaborated the description
- 71 in the revised manuscript: "There is a negative correlation between  $k_{total}$  and
- temperatures over 240 340 K, attributable to the presence of the pre-reactive
- 73 complexes (Giri et al., 2022; Chen et al., 2022) and the submerged TSs ( $\Delta E_a^{\sharp} < 0$ )
- 74 (Zádor et al., 2009) in the OH-addition pathway." (please see lines 145-147).

- 76 4. How sensitive are the calculated rate constants to variations in environmental
- 77 parameters such as temperature, pressure, or the concentration of OH radical? An
- 78 exploration of these dependencies under atmospherically relevant conditions would
- 79 strengthen the applicability of the findings.
- 80 Response: Thank you for the referee's helpful comment. According to the kinetic
- 81 calculation formula provided in the Supplement, the calculated rate constants are
- strongly sensitive to temperature, so the influence of temperature on rate constants has
- 83 been discussed in the revised manuscript: "There is a negative correlation between
- 84  $k_{\text{total}}$  and temperatures over 240 340 K, ...". (please see line 145); "At 340 K, the
- 85 OH-addition pathway accounts for 98.4% of the total reaction, this fraction
- 86 increases to 99.8% when the temperature is lowered to 240 K." (please see lines
- 87 155-156).
- 88 On the other hand, the atmospheric lifetime of BU is highly correlated with [OH], and
- thus, the discussion has been added in the revised version: "As shown in Table S5, the
- $\tau$  values are 9.29 and 5.93 hours for cis-BU and trans-BU, respectively, under the
- 91 remote areas with the [OH] of  $8 \times 10^5$  molecule cm-3, which are higher than those
- 92 of the corresponding lifetimes under the petrochemical regions with the [OH] of 1
- 93  $\times$  107 molecule cm-3." (please see lines 220-223).

- 95 5. According to Chen et al. (doi.org/10.1016/j.atmosenv.2020.118010), for 1-butene
- oxidation under high-NOx conditions, the  $RO_2 + NO$  pathway partitions roughly
- 97 equally ( $\approx$ 50 % each) to alkoxy radicals and to HCHO + HONO formation. Could the
- 98 authors quantify the analogous HONO yield from 2-butene and discuss why it appears
- 99 negligible in their mechanism, or provide computational/experimental evidence
- supporting this difference?
- Response: According to the referee's helpful comment, the yield of HONO formation

102 from BU has been quantified, and some discussions have been added to explain why 103 HONO formation can be neglected in the revised manuscript: "The yield of HONO is 104 calculated to be less than 1%, indicating the negligible formation (Table S9), in 105 contrast to the yield of 50% for 1-butene under high-NOx conditions reported by 106 Chen et al., (Chen et al., 2020). The discrepancy is attributed to the characteristic 107 of structure, i.e., the distinct  $\beta$ -hydrogen availability of the two alkenes." (please 108 see lines 181-184). 109 110 6. It is proposed that alkoxyl radicals (cis-RO and trans-RO) decompose into one 111 CH3CHO and one CH3CHOH radical, whereas the Master Chemical Mechanism 112 (MCM v3.3.1) assumes the direct cleavage yielding two CH3CHO. Please supply the 113 transition-state geometry and corresponding barrier for the "two CH3CHO" pathway, 114 and compare its rate constant with the CH3CHO + CH3CHOH pathway to justify why 115 the latter is favored under the studied conditions. Response: Thank you for the referee's helpful comment. We have included the relevant 116 117 kinetic data and optimized structures in Figure S9 and Table S13 of the Supplement. 118 Following the suggestion of Reviewer #2, we have updated the decomposition 119 mechanism of the alkoxy radicals (cis-RO and trans-RO) to a step-wise pathway 120 instead of the original two-step process. Therefore, we have compared using the 121 updated mechanism to justify the importance of the CH3CHOH radical pathway under 122 the studied conditions in the revised version: "According to the previous studies 123 (Zádor et al., 2009; Da Silva et al., 2009), CH3CHOH·radical (ER) undergoes a 124 step-wise O2-addition/HO2-elimination mechanism (R16) via ER-O2 formation, to 125 form CH3CHO and HO2 radical. The formation of ER-O2 is an exothermic process with the  $\Delta E_r$  value of -34.11 kcal mol-1, which supports to overcome the  $\Delta E_a^{\#}$  value 126 of 11.27 kcal mol-1 (Figures 4 and S9). However, under the petrochemical 127 128 conditions, ER-O2 readily react with NO, which is abundant in polluted areas, 129 forming ER-O2NO, since this reaction is a barrierless and largely exothermic with 130 the  $\Delta E$  value of -24.16 kcal mol-1 (R17)." (please see lines 200-205); "To elucidate 131 the significance of acid formation pathway in petrochemical regions, the kinetic 132 data were investigated and listed in Table S13. The rate constant for CH3CHO and HO2 formation is  $2.03 \times 10^{-11}$  cm3 molecule-1 s-1 at 298 K, which is slightly lower 133 than that for ER-O and NO2 formation (4.79 × 10-11 cm3 molecule-1 s-1)." (please 134

135

see lines 209-211).

Figure S9, PESs for the subsequent reaction of ER radical in predicted at the DLPNO-CCSD(T)/aug-cc-pVTZ//M06-2X/6-311+G(2df,2p) level of theory (in kcal mol-1).

Table S13. Rate constants of the reaction of R16(ER +  $O_2 \rightarrow AD + HO_2$ ), R16b(ER +  $O_2 \rightarrow ER-O_2$ ), R17(ER- $O_2 + NO \rightarrow ER-O_2NO + NO_2$ ), R18(ER- $O_2NO \rightarrow ER-O + NO_2$ ), R19(ER- $O_2 \rightarrow AA + HO_2$ ) computed at different temperatures (s-1 for R18, cm3 molecules-1 s-1 for R16, R16b, R17, and R19).

| T/K | R16                    | R16b                   | R17                    | R18                   | R19                    |
|-----|------------------------|------------------------|------------------------|-----------------------|------------------------|
| 240 | 2.49×10 -11 | 3.36×10 -11 | 4.30×10 -11 | 6.59×10 -1 | 1.01×10 -17 |
| 250 | 2.39×10 -11 | 3.43×10 -11 | 4.39×10 -11 | 2.12                  | 1.26×10 -17 |
| 260 | 2.3×10 -11  | 3.49×10 -11 | 4.47×10 -11 | 6.26                  | $1.56 \times 10^{-17}$ |
| 273 | 2.2×10 -11  | 3.58×10 -11 | 4.58×10 -11 | $2.27\times10^{1}$    | 2.05×10 -17 |
| 280 | 2.14×10 -11 | 3.63×10 -11 | 4.64×10 -11 | $4.32 \times 10^{1}$  | 2.36×10 -17 |
| 290 | 2.08×10 -11 | 3.69×10 -11 | 4.72×10 -11 | $1.03 \times 10^{2}$  | 2.87×10 -17 |
| 298 | 2.03×10 -11 | $3.74 \times 10^{-11}$ | 4.79×10 -11 | $1.97 \times 10^{2}$  | $3.34 \times 10^{-17}$ |
| 300 | 2.02×10 -11 | 3.75×10 -11 | 4.81×10 -11 | $2.31 \times 10^{2}$  | $3.47 \times 10^{-17}$ |
| 310 | 1.96×10 -11 | $3.81 \times 10^{-11}$ | 4.88×10 -11 | $4.91 \times 10^{2}$  | 4.16×10 -17 |
| 320 | 1.91×10 -11 | 3.88×10 -11 | 4.96×10 -11 | $1.00 \times 10^{3}$  | 4.97×10 -17 |
| 330 | 1.86×10 -11 | 3.94×10 -11 | 5.04×10 -11 | $1.95 \times 10^{3}$  | 5.89×10 -17 |
| 340 | 1.82×10 -11 | 3.99×10 -11 | 5.12×10 -11 | $3.65 \times 10^3$    | 6.95×10 -17 |

```
144
```

- 145 Minor comments:
- 146 Line 71: "reactants" should be "Rs";
- Response: "**reactants**" has corrected into "**Rs**" (please see line 75).
- 148 *Line 79: "Frisch 2009" should be "Frisch et al. 2009";*
- Response: "Firsch 2009" has corrected into "Firsch el al., 2009" (please see line 82).
- 150 Line 83: "tunneling" should be "tunnelling";
- Response: "tunneling" has corrected into "tunnelling" (please see line 97).
- Line 100: "the optimization" should be "the optimized structure";
- 153 Response: "the optimization" has corrected into "the optimized of structure" (please
- 154 see line 113).
- Line 115: Table S1 appears after Table S12 (Line 92), and Tables S2, S3 and S4 do not
- 156 *appear in the text;*
- 157 Response: "Table S12" has corrected into "Table S1" (please see line 115), and
- supplied Tables S2-S17 in the manuscript.
- 159 Line 120: "trans-BU" should be "that of trans-BU";
- Response: "trans-BU" has corrected into "that of trans-BU" (please see line 133).
- 161 Line 122: the subscript of "cis-TS" should be "cis-TSadd1";
- Response: "cis-TS" has corrected into "cis-TSadd1" (please see line 138).
- 163 Line 134: The reference "Sims et al., 1994" does not match the citation "Atkinson,
- 164 *2000"* in Figure 2;
- Response: "Sims et al., 1994" has corrected into "Atkinson, 2000" (please see line
- 166 140).

- 167 Line 214: "Atchem-2" should be "Atchem 2". Please check and revise accordingly.
- Response: "Atchem-2" has corrected into "Atchem2" (please see line 231).

**Referee #2**

170

- 171 The work presented in the manuscript consists of a theoretical reaction kinetics 172 investigation of the atmospheric photo-oxidation of 2-butene, an important 173 petrochemical volatile organic compound in the atmosphere. The authors report a 174 detailed mechanistic exploration, which is well depicted in the form of potential energy 175 diagrams. The manuscript also discusses the atmospheric implications of their findings, 176 making use of box model simulations. The computational methods employed in the 177 evaluated work are adequate, and the text is clear and concise, which is commendable. 178 However, there is one major issue to be addressed that potentially changes the main 179 message of the manuscript, that is, the high yield of organic acids from petrochemical 180 alkenes. I therefore recommend the reviewed manuscript to be reconsidered for
- 182 Response: We appreciate referee's suggestive comments.

publication after this major issue is addressed.

183

181

1. Page 7-8: "According to the previous study (Parandaman et al., 2018), 184 185 CH3CHOH·radical undergoes the H-abstraction pathway (R16) to form CH3CHO and 186  $HO_2$  radical. However, the corresponding  $\Delta E \# a$  and  $\Delta Er$  values are calculated to be 11.27 kcal mol-1 and 11.69 kcal mol-1, respectively, and the pathway of  $O_2$ -addition to 187 CH3CHOH radical (R17) is barrierless and largely exothermic (-34.11 kcal mol-1) to 188 189 yield peroxy radical (ER-O2) (Figure 4). More importantly, the subsequent NO-190 association of ER-O2 (R18) is also barrierless and largely exothermic (-24.16 kcal mol- 191 1). Therefore, peroxy nitrite (ER- $O_2NO$ ) is the dominant product of the subsequent 192 reaction of CH3CHOH via R17 and R18 rather than CH3CHO and HO2 radical via R16." 193

194195

196

197

198

199

200

201

202

203

The authors talk about a H-abstraction pathway (R16), which I assume to be the one involving direct abstraction by  $O_2$  (according to Figure 4), and reference an energy barrier value for calculated for it by Parandaman et al. (2018). However, the reaction investigated in the cited work is a concerted elimination reaction (where the H atom transfer occurs alongside cleavage of the C-O(O) bond), whose transition state is connected to the peroxyl radical, and not to the alkyl radical +  $O_2$  as assumed by the authors. This means that formation of acetaldehyde +  $HO_2$  can proceed via a step-wise addition/elimination mechanism ( $CH_3CHOH + O_2 \rightarrow E_2-RO_2 \rightarrow CH_3CHO + HO_2$ ), in which case an energy barrier of 11 kcal/mol associated with the second step is actually

quite low (meaning that the reaction is very fast). Furthermore, Parandaman et al. (2018) report calculations done only for geminal diols, and not for the specific system investigated in the reviewed work. That being said, the reaction of  $O_2$  with alphahydroxyethyl radical has already been studied previously by theory and experiments: See e.g. da Silva et al., 2009 (https://doi.org/10.1021/jp903210a) and Zádor et al., 2009 (https://doi.org/10.1016/j.proci.2008.05.020). According to their results, the energy barrier to the  $HO_2$ -elimination reaction is also quite low for  $E_2$ - $RO_2$  (11.4 kcal/mol), which I must say is a typical observation for this type of reaction pathway. On top of that, formation of  $E_2$ - $RO_2$  is 34 kcal/mol downhill in energy, so that the barrier to  $HO_2$ -elimination is submerged under the entrance energy level (see Fig 1a in https://doi.org/10.1016/j.proci.2008.05.020) and the overall reaction rate may be enhanced due to chemical activation.

My point here is that, while I agree with the authors that addition will almost certainly outcompete direct H-abstraction during the initial attack of  $O_2$  on  $CH_3CHOH$ , I argue that the subsequently formed peroxyl radical  $E_2$ -RO $_2$  will (very likely) react via the concerted elimination mechanism to yield acetaldehyde + HO $_2$  so rapidly that it outcompetes reaction with NO even at extremely high  $NO_x$  levels, so that formation of acetic acid via the route shown in Figure 4 becomes a negligible channel. The authors should then calculate the rate coefficient for this reaction, or at least take already calculated values from references mentioned above, and include it in their box model simulations. If, with the inclusion of this elimination reaction, the acetic acid forming channel turns out to be indeed minor, then the following sections may need to be reworked around the new results.

Response: Thank you for the referee's helpful comment. According to the referee's suggestion, the formation of acetaldehyde + HO2 has been treated as a step-wise addition/elimination mechanism in the revised manuscript: "According to the previous studies (Zádor et al., 2009; Da Silva et al., 2009), CH3CHOH·radical (ER) undergoes a step-wise O2-addition/HO2-elimination mechanism (R16) via ER-O2 formation, to form CH3CHO and HO2 radical. The formation of ER-O2 is an exothermic process with the  $\Delta E_r$  value of -34.11 kcal mol-1, which supports to overcome the  $\Delta E_a^{\mu}$  value of 11.27 kcal mol-1 (Figures 4 and S9). However, under the petrochemical conditions, ER-O2 readily react with NO, which is abundant in polluted areas, forming ER-O2NO, since this reaction is a barrierless and largely

exothermic with the ΔE value of -24.16 kcal mol-1 (R17)." (please see lines 200-205). According to the mechanism mentioned above, the kinetics including rate constants and branching ratios have been updated. The updated data shows that, when a step-wise addition/elimination mechanism is used, the branching ratio for CH3COOH formation remains as high as 70% at typical NO concentrations. Therefore, the relevant discussions have been updated in Sections 3.3: "To elucidate the significance of acid formation pathway in petrochemical regions, the kinetic data were investigated and listed in Table S13. The rate constant for CH3CHO and HO2 formation is 2.03 × 10-11 cm3 molecule-1 s-1 at 298 K, which is slightly lower than that for ER-O and NO2 formation (4.79 × 10-11 cm3 molecule-1 s-1). Furthermore, the predicted branching ratios forming CH3CHO and CH3COOH are 30% and 70%, respectively, which requires to further assess the contribution of BU to organic acids in the atmosphere." (please see lines 208-212).

Figure S9, PESs for the subsequent reaction of ER radical in predicted at the DLPNO-CCSD(T)/aug-cc-pVTZ//M06-2X/6-311+G(2df,2p) level of theory (in kcal mol-1).

Table S13. Rate constants of the reaction of R16(ER + O2  $\rightarrow$  AD + HO2), R16b(ER + O2  $\rightarrow$  ER-O2), R17(ER-O2 + NO  $\rightarrow$  ER-O2NO + NO2), R18(ER-O2NO  $\rightarrow$  ER-O + NO2), R19(ER-O + O2  $\rightarrow$  AA + HO2) computed at different temperatures (s-1 for R18, cm3 molecules-1 s-1 for R16, R16b, R17, and R19).

| T/K | R16                    | R16b                   | R17                    | R18                   | R19                    |
|-----|------------------------|------------------------|------------------------|-----------------------|------------------------|
| 240 | 2.49×10 -11 | 3.36×10 -11 | 4.30×10 -11 | 6.59×10 -1 | 1.01×10 -17 |
| 250 | 2.39×10 -11 | 3.43×10 -11 | 4.39×10 -11 | 2.12                  | 1.26×10 -17 |
| 260 | 2.3×10 -11  | $3.49 \times 10^{-11}$ | 4.47×10 -11 | 6.26                  | $1.56 \times 10^{-17}$ |
| 273 | 2.2×10 -11  | 3.58×10 -11 | 4.58×10 -11 | $2.27 \times 10^{1}$  | 2.05×10 -17 |
| 280 | 2.14×10 -11 | 3.63×10 -11 | 4.64×10 -11 | $4.32 \times 10^{1}$  | 2.36×10 -17 |
| 290 | 2.08×10 -11 | $3.69 \times 10^{-11}$ | 4.72×10 -11 | $1.03 \times 10^{2}$  | $2.87 \times 10^{-17}$ |
| 298 | 2.03×10 -11 | $3.74 \times 10^{-11}$ | 4.79×10 -11 | $1.97 \times 10^{2}$  | $3.34 \times 10^{-17}$ |
| 300 | 2.02×10 -11 | 3.75×10 -11 | 4.81×10 -11 | $2.31 \times 10^{2}$  | $3.47 \times 10^{-17}$ |
| 310 | 1.96×10 -11 | 3.81×10 -11 | 4.88×10 -11 | $4.91 \times 10^{2}$  | 4.16×10 -17 |
| 320 | 1.91×10 -11 | 3.88×10 -11 | 4.96×10 -11 | $1.00 \times 10^{3}$  | 4.97×10 -17 |
| 330 | 1.86×10 -11 | $3.94 \times 10^{-11}$ | 5.04×10 -11 | $1.95 \times 10^{3}$  | 5.89×10 -17 |
| 340 | 1.82×10 -11 | 3.99×10 -11 | 5.12×10 -11 | $3.65 \times 10^3$    | 6.95×10 -17 |

260 Minor comments:

259

261

262

263

264

265

266

267

268

269

270

271

272

273

274

275

276

277

278

2. Page 2, paragraph 2: The authors discuss the importance of the atmospheric oxidation of alkenes to air quality and climate in comparison to that of isoprene. However, important biogenic volatile organic compounds in the atmosphere, such as isoprene and most monoterpenes, are themselves also alkenes. I suggest that the authors include a mention to this fact in their introduction to further emphasize that their work is focused on an alkene from petrochemical origin, rather than biogenic. Response: According to the referee's helpful comment, an explanation has been added in the introduction to emphasize the importance of biogenic alkenes oxidation in the atmosphere and to highlight our focus on the oxidation of petrochemical-derived alkenes: "Previous studies have shown that multi-generation oxidation of biogenic and anthropogenic alkenes is important to the formation of local free radicals, tropospheric ozone, and SOA (Wu et al., 2021; Wang et al., 2022; Yang et al., 2023; Tan et al., 2024; Huang et al., 2025)" (please see lines 38-40); "The gas-phase oxidation of biogenic alkenes (isoprene and monoterpenes) produces abundant semi-volatile organic products, whose second- or later-generation products are major contributors to SOA (Wu et al., 2021)." (please see lines 41-43); "Recent

studies have pointed out that multi-generation oxidation of anthropogenic alkenes

contribute to 8 - 20% of SOA mass (Lee et al., 2022) and about 89% of O3

- formation (Yang et al., 2024) in the petrochemical regions." (please see lines 43-45);
- 280 "Given the ubiquity of anthropogenic alkenes in petrochemical areas,
- 281 understanding the multi-generation oxidation mechanisms of anthropogenic
- alkenes is crucial for accurately predicting their impacts on air quality." (please
- 283 see lines 46-48).
- 284
- 285 3. Page 2, line 40: "For example, the reaction of isoprene with nitrate radicals (NO3)
- 286 produces the high nitrogen-containing monomers and dimers, ...". What do the authors
- 287 mean by "high nitrogen-containing monomers and dimers"? Please rewrite for clarity.
- 288 Response: We thank the referee for this helpful suggestion. We have rewritten the
- sentence for clarity: "For example, the reaction of biogenic alkene, isoprene, with
- 290 nitrate radicals (NO3) produces some N-containing monomers and dimers, leading
- to an estimated yield of organic aerosol mass of approximately (5  $\pm$  2) % (Wu et
- 292 **al., 2021).**" (please see lines 40-41).
- 293
- 294 4. Page 3, line 69: "Harmonic vibrational frequencies were performed at the same level
- 295 to verify the nature of transition state (NIMAG=1) and minimum (NIMAG=0), ..."
- 296 I understand that "NIMAG" stands for "number of imaginary frequencies", however, a
- 297 reader who is less familiar with the methods may be confused with the acronym. Please
- write out what is meant with it.
- 299 Response: Thank you for the referee's helpful comment. We have revised the
- 300 corresponding sentence: "Harmonic vibrational frequencies were performed at the
- 301 same level to check all stationary points either a TS (with only one imaginary
- frequency) or the minima (without any imaginary frequencies), ..." (please see
- 303 lines 72-73).
- 304
- 5. Page 4, Line 100: "the optimization of geometries for all single points (SPs) involved
- 306 in these two reactions"
- 307 Do the authors mean "stationary points"?
- Response: According to the referee's suggestion, the mistake has been corrected in the
- 309 revised manuscript.
- 310
- 311 6. Figure 2: The experimental data shown with light blue and light pink overlaps with
- 312 theoretical data and is not that easy to see. I suggest using a different color scheme for

those data points to aid visualization.

Response: We appreciate the referee's suggestion. We have replaced the light-blue and light-pink symbols with blue crosses and red diamonds, respectively, for those data points to aid visualization, as shown in the revised Figure 2.

Figure 2: The rate constants (cm3 molecule-1 s-1) of (a) *cis*-BU and (b) *trans*-BU with OH against the temperature range of 240-340 K. The experimental rate constants are from Atkinson, 2000 and Ohta, 1984 respectively.

7. Page5, lines 131-132: "There is a negative correlation between rate constants and temperatures in the temperature range of 240 - 340 K, attributable to the presence of the pre-reactive complexes."

I would say that the negative correlation is due to the presence of a submerged reaction energy barrier rather than to the presence of a pre-reactive complex alone. Note that the transition state of OH-addition pathways, which dominate the overall rate coefficients, are lower in energy than the respective reactants. The rate coefficients of H-abstraction pathways, however, do not have submerged energy barriers and thus display positive temperature dependences (according to Figure 2).

Response: We thank the referee for this insightful comment. In original manuscript, according to the previous studies (Giri et al., 2022, *Phys. Chem. Chem. Phys.*, 2022, 24, 7836-7847; Chen et al., 2022, *J. Phys. Chem. A* 2022, 126, 19, 2976–2988), we stated that the negative temperature dependence is related to the pre-reactive complex. Combined with the reviewer's suggestion and our results, we agreed that the negative temperature dependence also originates primarily from the submerged energy barriers

- of the OH-addition pathways. Accordingly, the corresponding sentence have been
- revised: "There is a negative correlation between  $k_{\text{total}}$  and temperatures over 240
- 340 K, attributable to the presence of the pre-reactive complexes (Giri et al., 2022;
- Chen et al., 2022) and the submerged TSs ( $\Delta E_a^{\#} < 0$ ) (Zádor et al., 2009) in the OH-
- addition pathway." (please see lines 145-147).

- 8. Page6, line 167: "undergoes dissociation (cis-R13), isomerization (cis-R14), and H-
- 344 abstraction (cis-R15) yield acetaldehyde"
- 345 missing word: "... to yield .."
- Response: Thank you for the referee's helpful comment. We have revised in the
- manuscript as follows: "...undergoes dissociation (cis-R13), isomerization (cis-R14),
- and H-abstraction (cis-R15) to yield acetaldehyde (CH3CHO) and hydroxyalkyl
- 349 radicals (CH3CHOH), CH3CH(OH)CH(OH)CH2 radical, and acetoin
- 350 (CH3C(=O)CH(OH)CH3), respectively." (please see lines 184-186).